# The Decay of Pertussis Antibodies in Children Aged 0–14 Years in Jiangsu Province, China

**DOI:** 10.3390/vaccines11081336

**Published:** 2023-08-07

**Authors:** Wen Wang, Zhiguo Wang, Qiang Chen, Mei Li, Chengmei Jia, Yan Xu, Yun Wu, Xiang Sun, Hui Sun

**Affiliations:** 1Department of Rheumatology and Immunology, The Affiliated Suqian First People’s Hospital of Nanjing Medical University, Suqian 223800, China; 2Department of Expanded Program on Immunization, Jiangsu Provincial Center for Disease Control and Prevention, Nanjing 210009, China; wangzhiguo@jscdc.cn (Z.W.);; 3Medical Department, Affiliated Hospital of Nanjing University of Traditional Chinese Medicine, Nanjing 210000, China

**Keywords:** pertussis, restricted cubic spline, incidence, anti-PT IgG, anti-FHA IgG

## Abstract

The purpose of this study was to investigate possible influencing factors based on the distribution of the pertussis toxin (PT) and filamentous hemagglutinin (FHA) antibody levels in 0–14-year-old children in Jiangsu Province, China, and to analyze changes in IgG antibody levels after pertussis vaccination in children over time via a restricted cubic spline (RCS)-fitted binary logistic regression model. We collected surveillance data on pertussis through the National Notifiable Disease Reporting System (NNDRS). Serum samples were collected, and PT IgG/FHA IgG antibody levels were determined via an enzyme-linked immunosorbent assay (ELISA). A binary logistic regression model was fitted with an RCS. Peak incidence occurred in children aged 0–1 years from 2007 to 2022, and a second peak emerged in children aged 5 years and older from 2018 onwards which shifted towards older age groups. The geometric mean concentrations (GMC) of the anti-PT IgG antibody and anti-FHA IgG antibody in 1129 patients were 15.13 (13.49–16.76) IU/mL and 22.99 (21.17–24.81) IU/mL, respectively. The seropositivity rates of the anti-PT IgG and anti-FHA IgG antibodies in the group receiving a full vaccination course (four doses) were significantly higher than those of other groups (24.6% vs. 43.3%). The RCS fitting model showed a non-linear relationship between the duration after immunization and the odds ratio (OR) of having PT-IgG and FHA-IgG antibody concentrations ≥20 IU/mL in children with documented immunization histories (1–4 doses) (P_overall_ < 0. 001; P_nonlinear_ ≤ 0.001). The children with histories of immunization demonstrated antibody levels that decreased to very low levels around 17 months after the last dose of the vaccine. Therefore, it is recommended that pertussis-containing vaccines be administered as booster immunizations for older children.

## 1. Introduction

Pertussis, commonly referred to as whooping cough, is a highly contagious illness caused by the bacteria Bordetella pertussis (Bp). The introduction of widespread pertussis vaccination programs since 1974 has led to a substantial reduction in the incidence and mortality rates associated with the disease. In China, the diphtheria, tetanus, and whole-cell pertussis combined vaccine (DTwP) was incorporated into the national immunization program in 1978. In the mid-1990s, the acellular pertussis vaccine (DTaP) was developed and has been increasingly produced ever since [1]. DTaP offers several advantages over the previously used DTwP. The preparation of DTaP adopts co-purification technology in which PT and FHA are the main components, which are responsible for generating the immune response. This formulation reduces the risk of adverse reactions associated with the whole-cell vaccine while maintaining its protective efficacy against pertussis. As a result, DTaP has become the preferred vaccine in many countries, including China, due to its improved safety profile.

In 2007, the DTaP was introduced into the expanded national immunization program in China. Subsequently, in 2012, it replaced the previously used DTwP. The DTaP vaccine is administered on a three-dose schedule, with doses provided at 3, 4, and 5 months of age, followed by a booster dose at 18 months. The implementation of DTaP vaccination has had a significant impact on pertussis incidence in China. Prior to the introduction of the vaccine, the incidence of pertussis ranged from 100 to 200 cases per 100,000 individuals. However, following the widespread use of DTaP, the incidence has drastically decreased to approximately 1 case per 100,000 people [2]. However, despite the increasing coverage of pertussis vaccination, many countries have reported a resurgence of the disease in recent years, regardless of the type of vaccine used (whole-cell or acellular) or the vaccination schedule [3,4,5,6]. In America, California experienced its largest pertussis outbreak in more than 50 years in 2010–2011, followed by an even larger outbreak in 2014–2015 [7]. Similarly, the incidence of pertussis in China has also increased in recent years, with more than 30,000 reported cases in 2019, according to the National Notifiable Infectious Diseases Surveillance System (NNDSS) [8]. In 2022, Jiangsu Province reported 853 cases of pertussis, an 81.24% increase compared to the previous year (Figure 1). Since the 1990s, children under 1 year of age have represented the main demographic presenting pertussis, and there has been a dramatic increase in the number of reported cases in children aged 7–14 years in the 21st century [2].

The global resurgence of pertussis is a complex phenomenon influenced by various factors. Among these factors, improvements in laboratory testing technologies and increases in awareness among clinicians for accurate diagnosis have contributed to the identification and reporting of more pertussis cases. However, one of the central factors contributing to this resurgence is the declining effectiveness of the DTaP vaccine (although both whole-cell and acellular pertussis vaccines are effective, the acellular vaccine may have somewhat lower efficacy in preventing pertussis compared to the whole-cell vaccine; this reduced effectiveness, combined with waning immunity, could contribute to breakthrough infections), as highlighted in multiple references [7,9,10,11]. Following pertussis vaccination, children experience a gradual decline in antibody levels over time. Research studies have consistently demonstrated that approximately one year after vaccination, specific antibody levels in children rapidly decrease below the protective threshold [12]. This decline in antibody levels suggests that the immunity conferred by the DTaP vaccine wanes over time, making individuals more susceptible to pertussis infection. The serum titers of pertussis IgG antibodies against PT and FHA are assessed to monitor vaccine-induced herd immunity. However, recent sero-epidemiological studies on pertussis have shown that antibody levels decline after vaccination, and there is an urgent need to strengthen dosages in the Chinese population. It is important to note that observing the continuous change in the trend of pertussis antibody levels in children over time is challenging due to various factors, including the influence of natural infection and other confounding variables. Therefore, in many cross-sectional studies, antibody levels for pertussis in children aged 0–14 years only show a decreasing trend [13,14].

RCS analysis is a statistical method that combines elements of polynomial regression and segmented regression. It is commonly employed to examine non-linear relationships between continuous independent variables and dependent variables, offering a means to visualize the dose–response relationship [15]. In the context of this study, the aim is to explore potential influencing factors based on the distribution of PT and FHA antibody levels in children aged 0–14 years in Jiangsu Province. This study utilizes a binary logistic regression model with RCS fitting to analyze changes in IgG antibody levels over time following pertussis vaccination in children. The ultimate goal of this research is to enhance the immunization program for pertussis vaccines within the country and provide insights to improve the prevention and control of pertussis.

## 2. Methods

### 2.1. Pertussis Surveillance

To ensure the comprehensive and accurate reporting of pertussis cases, it is mandatory for hospital staff to record all diagnosed cases in the NNDRS. The NNDRS is a web-based computerized reporting system specifically designed for this purpose. By utilizing this system, healthcare professionals can efficiently report and monitor pertussis cases, enabling timely and effective disease surveillance and control measures. The incidence rate of pertussis is calculated as the number of cases per 100,000 individuals. The population denominator used for this calculation is provided by the National Bureau of Statistics of China. This denominator represents the total population at risk and serves as an essential factor in determining the burden of pertussis within the population. The utilization of the NNDRS and the calculation of the incidence rate using the population denominator provided by the National Bureau of Statistics of China contribute to the systematic and standardized monitoring of pertussis cases across the country.

### 2.2. Serological Survey

All serum samples were collected from healthy individuals who participated in the Jiangsu Provincial Health Population Vaccine Antibody Level Monitoring Project from 2019 to 2022. This monitoring project was conducted in four cities in Jiangsu Province, including Suqian, Yancheng, Changzhou, and Huai’an. The recruitment of participants was carried out by sending WeChat notifications to the parents of children aged 0–14 years who met the inclusion criteria, which involved being in good health and not currently experiencing acute pertussis or recent infection. Prior to enrollment, each participant underwent a medical examination and health assessment conducted by a qualified doctor. The study protocol was reviewed and approved by the Ethics Committee of the Jiangsu Provincial Center for Disease Control and Prevention, ensuring that ethical considerations were appropriately addressed. Written informed consent was obtained from either the individuals themselves or from the parents of the participating children. To gather relevant information, the participants were asked to complete an anonymous questionnaire which included details such as their date of birth, gender, place of residence, date of serum collection, and their pertussis immunization history.

### 2.3. Laboratory Assay

Serum samples were collected and stored at −20 °C until testing. An enzyme-linked immunosorbent assay (ELISA) was used to detect pertussis-specific IgG antibodies; all the experiments were performed at the laboratory of the Department of the Expanded Programme on Immunization at the Jiangsu Provincial Center for Disease Control and Prevention. To avoid test bias, all detection tests were performed by the same staff members using commercial ELISA kits from Zhengzhou Yit Bio-Tech Co., Ltd. (Zhengzhou, China), which was previously determined against NIBSC 06/140, and the antibody results obtained from the tests were expressed in international units per milliliter (IU/mL). According to the manufacturer’s instructions, the minimum level of anti-PT IgG detection was 5 IU/mL, and anti-PT IgG and anti-FHA IgG antibody concentrations <20 IU/mL were considered negative. If ≥20 IU/mL is judged positive, it is considered to provide immune protection to the human body. An anti-PT IgG concentration ≥80 IU/mL was considered to indicate a recent infection if the subject had not received the pertussis-containing vaccine within the previous year.

### 2.4. Statistical Analysis

The data analysis was performed utilizing several software tools: Microsoft Excel, R software (version 4.2.2), and SPSS V.22.0 (IBM Corp, Armonk, NY, USA). Geometric mean concentrations (GMCs) (the geometric mean summarizes the central tendency of the antibody concentrations in the sample, and it was calculated by multiplying all individual concentrations and then taking the nth root (*n* = number of individuals)) and the corresponding seropositivity rates of the antibodies (seropositivity was defined as detectable antibody levels above 20 IU/mL, and the seropositivity rate was calculated by dividing the number of seropositive individuals by the total sample size), along with their respective 95% confidence intervals (95% CI), were calculated and organized based on region, gender, and age. To compare the GMCs, an analysis of variance (ANOVA) was employed, considering a two-tailed *p*-value of 0.05 as the threshold for statistical significance. A two-segment logistic regression model was fitted using an RCS to explore the relationship between the duration of immunity after vaccination and PT-IgG and FHA-IgG antibody levels in children. The number of knots in the RCS fit was chosen based on the model’s maximum R^2^ to avoid overfitting. In the analysis, the x-axis represented the duration of immunity after vaccination, while the y-axis represented the OR value of the predicted PT-IgG and FHA-IgG antibody concentrations ≥20 IU/mL. The OR values were plotted, and the color intervals represented the corresponding 95% CI of the OR. The association trend was assessed by conducting both overall and non-linear Wald χ^2^ tests. If the overall Wald χ^2^ test resulted in a *p*-value of less than 0.05, indicating statistical significance, while the non-linear Wald χ^2^ test yielded a *p*-value greater than 0.05, a linear association trend between the variables being analyzed was suggested. On the other hand, if both the overall and non-linear Wald χ^2^ tests resulted in *p*-values less than 0.05, a non-linear association trend between the variables was indicated.

## 3. Results

### 3.1. Changes in the Epidemiological Characteristics of Pertussis (0–14 Year) in Jiangsu Province, 2007–2022

During the study period, we collected data on 1959 confirmed cases of pertussis among children aged 0–14 years. Our analysis revealed two distinct pertussis epidemics, one in 2019 and another in 2022 (Figure 1). These epidemic years exhibited higher levels of bacterial circulation compared to the intervening years. Notably, we observed two distinct age distribution patterns. Firstly, there was a consistent peak incidence among children aged 0–1 years from 2007 to 2022. Secondly, a second peak emerged in children aged 5 years and older from 2018 onwards, gradually shifting towards older age groups. It is interesting to highlight that in previous reports on pertussis cases, children aged 0–1 years constituted approximately 80% of the observed cases among children aged 0–14 years from 2007 to 2020. However, this proportion decreased to 37.21% (374 out of 1005 cases) during the period from 2021 to 2022. In contrast, the number of cases among children aged 5–14 years accounted for 47.56% (478 cases) during the same period. (Figure 2A,B).

### 3.2. The Basic Situation of the Study Population

A total of 1129 children participated in the study, with varying numbers of participants in different age groups ranging from 19 to 335 children in each group. A significant proportion of the children in the study did not receive the recommended pertussis vaccine dose, while others completed the full course of treatment with a booster dose. The immunization status of a small proportion of children could not be determined due to unknown or missing records. The details are as follows: 472 (41.81%) children did not receive a full course of immunization, usually consisting of 1–3 doses. On the other hand, 582 children (51.55%) completed the entire immunization course, including the booster dose, for a total of four doses. The immunization history of 56 children (4.96%) was unknown. (Table 1).

### 3.3. Serological Analysis

The GMC of the anti-PT IgG antibodies in the 1129 children aged 0–14 years old was measured at 15.13 (13.49–16.76) IU/mL. Table 1 provides additional insights into the seropositivity rates and GMC values across different age groups. The highest seropositivity rate (≥20 IU/mL) was observed in the 18–23-month age group at 31.6%, while the lowest rate was found in the 12–17-month age group at 12.7%. There was a notable decline in seropositivity rates from 31.6% in the 18–23-month age group to 13.5% in the 2–4-year age group, which then remained at relatively low levels in all other age groups. The analysis of anti-PT IgG antibody levels revealed interesting patterns across different age groups. The GMC of the anti-PT IgG antibody levels demonstrated a significant increase in the 0–2-months age group, followed by a subsequent decline. The first peak was observed in the 7–11-month age group, with a GMC value of 17.95 IU/mL. This was followed by a decline to 11.78 IU/mL in the 12–17-month age group. Subsequently, a second peak occurred in the 18–23 months age group, with a GMC value of 20.33 IU/mL. With the exception of the 0–2-month age group, the proportion of individuals with anti-PT IgG antibody levels equal to or greater than 80 IU/mL was highest in the 5–6-year age group (5.3%). In all other age groups, this proportion ranged from 2.0% to 3.9%. The GMC of the anti-FHA IgG antibody was 22.99 (21.17–24.81) IU/mL, and the seropositivity rate was 36.0%. The FHA antibody trend in different age groups was similar to that the PT trend, but with slight differences. The seropositivity rate and the GMCs of the anti-PT IgG and anti-FHA IgG antibodies were highest in the 18–23-month age group (31.6% vs. 48.7%; 20.33 vs. 28.26). In addition, the seropositivity rate of anti-FHA IgG antibodies among individuals who received the full course of vaccination (four doses) was significantly higher compared to the incomplete vaccination group (1–3 doses) and non-vaccinated group. (Table 1).

### 3.4. The Trend of PT-IgG and FHA-IgG Antibody Levels with the Time after Immunization

The RCS analysis revealed a non-linear relationship between the duration after immunization and the OR of having PT-IgG and FHA-IgG antibody concentrations ≥20 IU/mL in children with documented immunization history (1–4 doses) (P_overall_ < 0.001, P_nonlinear_ ≤ 0.001). The inflection points of the RCS curves occurred at 17.71 months (PT-IgG) and 16.95 months (FHA-IgG) after immunization, respectively, after which the OR of having antibody levels exhibited a gradually increasing and then decreasing non-linear trend (Figure 3A,B). In addition, we fitted a binary logistic regression model with ORs using RCS curves to explore the relationship between the duration after immunization and the OR of having PT-IgG and FHA-IgG antibody concentrations ≥20 IU/mL in children who had completed the full course of pertussis vaccination (four doses). The inflection points of the RCS curves occurred at 32.54 months (PT-IgG) and 31.29 months (FHA-IgG) after immunization, respectively (Figure 3C,D).

## 4. Discussion

The inclusion of the DTaP combination vaccine in China’s immunization program has resulted in significant advancements in the prevention and control of pertussis. Over the years, the incidence rate of pertussis has seen a substantial decrease. In the 1960s–1970s, the incidence rate ranged from 100 to 200 cases per 100,000 individuals [16]. However, in recent years, the incidence rate has dropped to approximately 1 case per 100,000 individuals [2,17].

Multiple factors have contributed to this decline in pertussis incidence. The introduction and widespread use of the DTaP vaccine have played a vital role in reducing the transmission of the bacteria responsible for pertussis. The vaccine’s immunization coverage, especially among infants and young children, has improved, resulting in a significant reduction in the overall burden of pertussis. Furthermore, in 2020, the strict prevention and control measures implemented during the COVID-19 pandemic may have contributed to the decline in pertussis transmission. Measures such as social distancing, mask-wearing, and enhanced hygiene practices could have indirectly impacted the spread of pertussis bacteria as they are transmitted through respiratory droplets. These measures likely played a role in reducing the overall transmission of respiratory pathogens, including the bacteria causing pertussis [18,19].

Despite a high DTaP vaccination rate, the incidence of pertussis has risen in China and other countries [20,21]. The DTaP vaccine aims to reduce severe cases and mortality in infants but has limited effectiveness in preventing infection or transmission. Studies show that DTaP mainly targets Th1/Th2 responses, stimulating specific immune cells and antibody production, reducing the severity of pertussis but offering limited and short-lived protection [22,23]. Although DTaP vaccines do provide protection for the first few years of life, changes in T cell responses lead to a decline in the effectiveness of acellular pertussis vaccines 2–3 years after booster shots, with immune persistence lasting around 5 years [24,25,26]. In addition to waning immunity, factors like improved clinical recognition, increased pertussis strain virulence, and better diagnosis contribute to more reported cases, raising pertussis awareness and identification.

Pertussis cases mostly occurred in infants younger than 1 year of age due to their susceptibility, clinical symptoms, and easy diagnosis. While PT-IgG antibodies can transfer from mother to fetus [27,28], China lacks an adult DTaP vaccine, leading to low maternal antibody levels. Thus, infants struggle to obtain sufficient protective antibodies [29]. However, this study discovered PT-IgG and FHA-IgG levels above 20 IU/mL in infants aged 0–2 months, revealing a severe underestimation of the incidence of pertussis in Chinese adults. In this study, the GMCs of anti-PT and anti-FHA antibodies significantly increased after primary immunization (at 3, 4, and 5 months of age) and booster immunization (at 18–24 months of age), and compared with children who completed the entire series of pertussis vaccination, children who did not complete the vaccination course had lower levels of anti-PT and anti-FHA antibodies (PT: 17.88 IU/mL vs. 11.19 IU/mL, FHA: 27.37 IU/mL vs. 16.76 IU/mL). Timely and effective vaccination helps prevent pertussis in these age groups, but over time, the levels of anti-PT and anti-FHA antibodies continue to decline; in particular, starting at 2 years of age, the GMC of anti-PT antibodies in more than 50% of children drops to below 5 IU/mL. The recent infection indicator PT-IgG ≥ 80 IU/mL also shows that 5.3% of children in the 5–6 age group may have been recently infected.

Previous studies have primarily focused on describing changes in the positivity rate or antibody concentration of pertussis vaccines following immunization, often providing simplistic explanations. In contrast, this study employs a more sophisticated approach by utilizing an RCS to examine the independent effect of post-immunization duration on antibody levels. By fitting a spline graph of binary logistic regression models using an RCS, we found a non-linear association trend between post-immunization duration in children with immunization history and OR values of PT-IgG and FHA-IgG antibody concentrations ≥20 IU/mL, indicating that the PT-IgG and FHA-IgG antibodies had dropped below protective levels around 17 months after vaccination (1–4 doses), while for children who completed the full course of immunization, the antibodies dropped to the lowest level at around 32 months. This finding further suggests that timely and effective full immunization leads to longer antibody duration. These results align with US and European studies on immunity decline [7,30,31,32]. From 2021 to 2022, children aged 5–14 accounted for 47.56% of the cases, with school-age children and adolescents being more active and hence major contributors to disease transmission, especially for unvaccinated infants who are a potential direct and indirect source of pertussis infection and at high risk of mortality. The American Advisory Committee on Immunization Practices (ACIP) recommends five pertussis vaccine doses for infants [33], while the Chinese Pertussis Action Plan suggests using DTaP for six-year-olds to improve population immunity [34]. However, unfortunately, vaccines for adolescents or adults have not been approved for use in China to date.

This study has several limitations that should be acknowledged. Firstly, it is important to note that this study was conducted as a cross-sectional analysis, which means that it captured a snapshot of data from children aged 0–14 years. As a result, it did not observe the longitudinal changes in antibody levels following immunization within a prospective cohort study in a real-world setting. Consequently, the findings should be interpreted with caution, as they may not fully reflect the dynamics of antibody levels over time. Furthermore, this study solely focused on analyzing the decay of antibodies at different time points post immunization. The absence of a comprehensive evaluation of other factors, such as vaccine efficacy or the presence of potential confounding variables, limits the depth of the conclusions that can be drawn from this study. Additionally, it is worth noting that the variability in testing kits from different manufacturers may have influenced the comparability of results across studies. Inconsistent standards and variations in assay performance could introduce a certain degree of bias or uncertainty when comparing findings between different research endeavors. Lastly, this study did not include information about the incidence of adult pertussis or the role of adult infections in the resurgence of pertussis. Further research is needed to investigate these aspects and provide more comprehensive insights into the dynamics of pertussis transmission.

In summary, this study indicates low levels of pertussis antibodies in children aged 0–14 years, with over 50% having levels below 5 IU/mL after age 2 and some being undetectable. The model fitting analysis revealed a significant decline in antibody levels among children who had received immunization, reaching very low levels approximately 17 months after the last dose of the DTaP vaccine. Administering pertussis-containing booster vaccines for older children is recommended to reinforce waning antibodies and ensure sustained protection. Further research and monitoring are crucial to validate findings and assess the booster immunization’s long-term impact on pertussis control.

## Figures and Tables

**Figure 1 vaccines-11-01336-f001:**
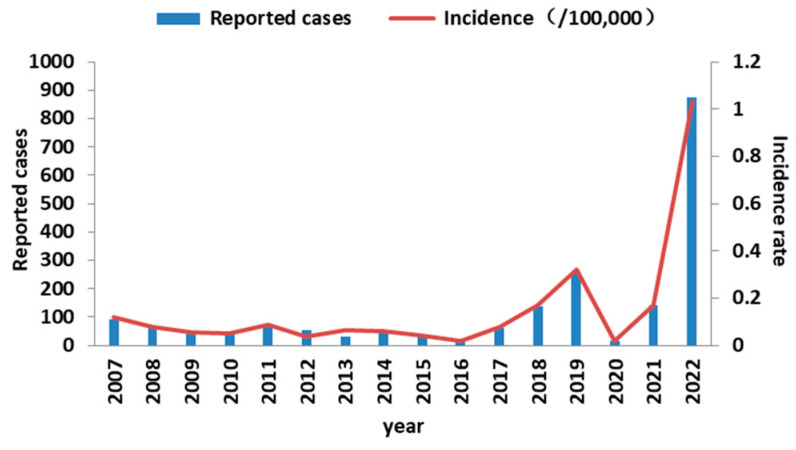
Reported cases and incidence from 2007–2022 in Jiangsu Province, China.

**Figure 2 vaccines-11-01336-f002:**
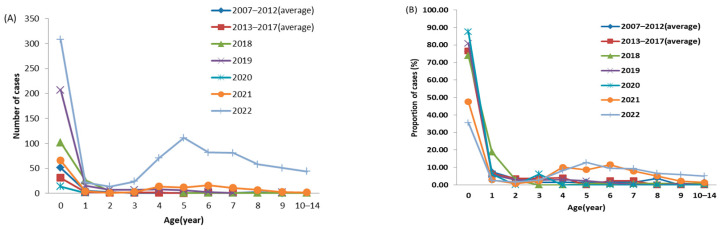
Cases of pertussis in people aged 0–14 years in Jiangsu, 2007–2022. (**A**) Number of cases by age and year. (**B**) The proportion of cases among individuals aged 0–14 years by age and year.

**Figure 3 vaccines-11-01336-f003:**
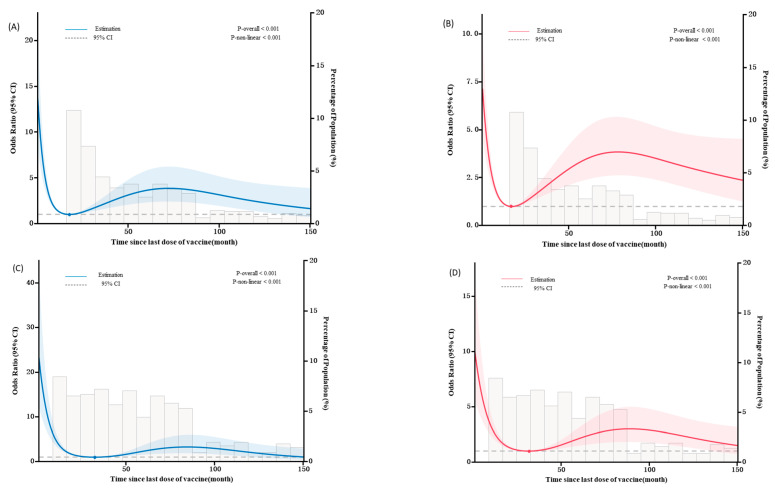
The duration after immunization was determined according to the minimum OR (OR = 1) of the RCS model fitting; the OR can reflect the relationship between the duration after immunization and the IgG antibody concentrations (≥20 IU/mL). A graph was then plotted with the post-immunization duration on the x-axis and the OR value for the model-predicted PT-IgG antibody concentration ≥20 IU/mL on the y-axis. The color intervals represent the 95% confidence interval for the OR. (**A**) The duration after immunization and the OR of having PT-IgG antibody concentrations ≥20 IU/mL in children with a documented immunization history (1–4 doses). (**B**) The duration after immunization and the OR of having FHA-IgG antibody concentrations ≥20 IU/mL in children with a documented immunization history (1–4 doses). (**C**) The duration after immunization and the OR of having PT-IgG antibody concentrations ≥20 IU/mL in children who received a full course of pertussis vaccination (four doses). (**D**) The duration after immunization and the OR of having FHA-IgG antibody concentrations ≥20 IU/mL in children who received a full course of pertussis vaccination (four doses).

**Table 1 vaccines-11-01336-t001:** Demographic data and distribution of anti-PT IgG and anti-FHA IgG concentrations in children 0–14 years of age.

Characteristic	*n*	Anti-PT IgG					Anti-FHA IgG			
		Positivity Rate (%)			Infection Rate (%)	GMC (95% CI) (IU/mL)	Positivity Rate (%)			GMC (95% CI) (IU/mL)
		<5 IU/mL	5–19 IU/mL	≥20 IU/mL	≥80 IU/mL		<5 IU/mL	5–19 IU/mL	≥20 IU/mL	
Age (month/year) *										
0–2 months	19	13 (68.4)	0 (0.0)	6 (31.6)	3 (15.8)	23.42 (4.87–41.96)	10 (52.6)	3 (15.8)	6 (31.6)	24.37 (7.08–41.65)
3–6 months	122	34 (27.9)	66 (54.1)	22 (18.0)	3 (2.5)	14.67 (10.57–18.77)	39 (32.0)	67 (54.9)	16 (13.1)	14.22 (10.03–18.42)
7–11 months	50	19 (38.0)	16 (32.0)	15 (30.0)	1 (2.0)	17.95 (9.59–26.31)	8 (16.0)	19 (38.0)	23 (46.0)	26.31 (16.88–35.75)
12–17 months	63	39 (61.9)	16 (25.4)	8 (12.7)	2 (3.2)	11.78 (6.42–17.15)	18 (28.6)	25 (39.7)	20 (31.7)	19.80 (12.89–26.71)
18–23 months	152	49 (32.2)	55 (36.2)	48 (31.6)	4 (2.6)	20.33 (16.34–24.33)	7 (4.6)	71 (46.7)	74 (48.7)	28.26 (24.02–32.53)
2–4 years	274	190 (69.3)	47 (17.2)	37 (13.5)	7 (2.6)	10.41 (7.62–13.20)	94 (34.3)	94 (34.3)	86 (31.4)	18.43 (14.97–21.90)
5–6 years	114	73 (64.0)	17 (14.9)	24 (21.1)	6 (5.3)	16.45 (10.28–22.62)	31 (27.2)	44 (38.6)	39 (34.2)	24.05 (17.77–30.32)
7–14 years	335	180 (53.7)	88 (26.3)	67 (20.0)	13 (3.9)	16.06 (12.63–19.53)	78 (23.3)	115 (34.3)	142 (42.4)	27.17 (23.39–30.94)
Sex										
Male	630	337 (53.5)	167 (26.5)	126 (20.0)	20 (3.2)	14.53 (12.57–16.48)	168 (26.7)	233 (37.0)	229 (36.3)	22.63 (20.38–24.87)
Female	499	260 (52.1)	138 (27.7)	101 (20.2)	19 (3.8)	15.88 (13.12–18.64)	117 (23.4)	205 (41.1)	177 (35.5)	23.44 (20.45–24.81)
Immunization history (doses) *										
0	19	13 (68.4)	0 (0.0)	6 (31.6)	3 (15.8)	23.42 (4.87–41.96)	10 (52.6)	3 (15.8)	6 (31.6)	24.37 (7.08–41.65)
1–3	472	249 (52.8)	157 (33.3)	68 (14.0)	7 (1.5)	11.19 (9.46–12.93)	121 (25.6)	229 (48.5)	122 (25.8)	16.76 (14.66–18.87)
4	582	298 (51.2)	141 (24.2)	143 (24.6)	25 (4.3)	17.88 (15.23–20.53)	138 (23.7)	192 (33.0)	252 (43.3)	27.37 (24.05–30.32)
Unknown	56	37 (66.1)	7 (12.5)	12 (21.4)	4 (7.1)	16.78 (8.25–25.32)	16 (28.6)	14 (25.0)	26 (46.4)	29.41 (20.15–38.68)
Registered population										
Native	948	489 (51.6)	264 (27.8)	195 (20.6)	33 (3.5)	15.26 (13.48–17.03)	240 (25.3)	365 (38.5)	343 (36.2)	22.95 (20.98–24.93)
Migrant	181	108 (59.7)	41 (22.7)	32 (17.7)	6 (3.3)	14.45 (10.19–18.70)	45 (24.9)	73 (40.3)	63 (34.8)	23.15 (18.38–27.91)
Total	1129	597 (52.9)	305 (27.0)	227 (20.1)	39 (3.5)	15.13 (13.49–16.76)	285 (25.2)	438 (38.8)	406 (36.0)	22.99 (21.17–24.81)

* The difference between groups was statistically significant, *p* < 0.001.

## Data Availability

The data that support the findings of this study are available upon request from the corresponding author. The data are not publicly available due to privacy or ethical restrictions.

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
